# Antibacterial Activity and Antibacterial Mechanism of Lemon Verbena Essential Oil

**DOI:** 10.3390/molecules28073102

**Published:** 2023-03-30

**Authors:** Xin Gao, Jinbao Liu, Bo Li, Jing Xie

**Affiliations:** 1College of Food and Tourism, Shanghai Urban Construction Vocational College, Shanghai 201415, China; gaoxin@succ.edu.cn (X.G.); liujinbao@succ.edu.cn (J.L.); 2College of Food Sciences and Technology, Shanghai Ocean University, Shanghai 200233, China

**Keywords:** lemon verbena essential oil, antibacterial activity, antibacterial mechanism, *Pseudosciaena crocea*

## Abstract

The destructive effect and mode of action of lemon verbena essential oil on cells were investigated, taking the isolated *Pseudosciaena D4* as the research object. The extracellular absorbance of the *Pseudosciaena D4* increased at OD260 and OD280 after being treated with lemon verbena essential oil, which destroyed the integrity of *Pseudosciaena D4* cells, showing a significant effect on preventing biomembrane formation and destroying the formed biomembrane. With an increased concentration of lemon verbena essential oil, extracellular polysaccharide showed a significant decrease in content and a significant increase in inhibition rate, indicating that the secretion of extracellular polysaccharide by *Pseudosciaena D4* cells could be inhibited by lemon verbena essential oil during the process of biomembrane formation. Cell introcession and shrinkage appeared after the treatment with essential oil, and a transparent cavity was formed by the out-flowed cell content. Lemon verbena essential oil destroyed the cell wall, resulting in an enhanced permeability of the cell membrane and leakage of the contents, thereby causing cell death.

## 1. Introduction

Marine fish are a popular dietary choice due to their rich nutrition, including protein and polyunsaturated fatty acids, as well as selenium, iodine, vitamin D, and other nutrients conducive to health [1,2]. The World Health Organization (WHO) recommends eating 1–2 meals of fish per week to provide 200–500 mg of omega-3-PUFA [3]. However, the quality of fresh fish is prone to decline after it is caught. The nutritional composition of most fish is degraded by digestive enzymes, lipases, and surface microorganisms [4]. In addition, changes in the composition of fish during decay lead not only to lipid oxidation and protein degradation, but also to the loss of other valuable molecules.

In the process of the refrigeration of aquatic products, *Pseudomonas*, a Gram-negative bacterium with simple nutritional needs, is considered to be a common specific spoilage organism (SSO) [5,6,7], which can be separated from aquatic products, livestock products, and dairy products [8]. *Pseudomonas* can resist harsh ambient conditions through defense mechanisms. The “stress response” appears in cases of exhausted nutrients or dramatically changing temperature so as to improve its ability to adapt to adverse environments depending on the products of gene-coordinated expression [9,10]. *Pseudomonas* are even able to cause keratitis in some cases [11]. To counteract the spoilage-causing effect of *Pseudomonas*, many researchers have gradually developed new green active packaging and natural biodegradable active films that inhibit specific spoilage bacteria such as *Pseudomonas* in the storage and transportation of fruit and vegetable foods in order to reduce economic and food distribution losses [12].

Essential oils are an important source of by-products from the agro-food processing industry (peels, seeds, and pulps) and are purified from the outer cortex glands of fruits and vegetables with strong biological activity. For example, citrus essential oils have strong antioxidant properties and antibacterial activity [13] and are widely used in products such as beverages, ice cream, cakes, air fresheners, household products, and perfumes [14]. Lemon verbena essential oil is a kind of plant essential oil extracted from lemon verbena. Its main component is citral, which has strong antioxidant and antibacterial abilities [15]. The plant and its essential oil have been widely used in different traditional medicines; for example, lemon verbena essential oil can inhibit the putrefactive effect of *Listeria monocytogenes* in beef [16] and can improve the stability of sunflower oil, which has been associated with the presence of flavonoids and phenolic acids [17]. However, the effect of lemon verbena essential oil on *Pseudomonas* has not been reported. Therefore, the previously isolated *Pseudomonas D4* was taken as the research object to investigate the destructive effect and mode of action of lemon verbena essential oil on its cells, so as to provide a basis for lemon verbena essential oil as a natural preservative in cold storage and preservation processes of aquatic products.

## 2. Materials and Methods

### 2.1. Bacterial Strain Pseudosciaena D4

Isolated strain *Pseudosciaena D4* (the similarity between D4 and *Pseudomonas* sp. strain U3-n-1 was 99.80% after 16 s identification): The bacteria were stored in 9 mL of liquid medium (broth MHB) (1% inoculum) and incubated in a rotatory shaker for 16 h to create the bacterial suspension for standby. According to the experimental grouping, enough bacteria were expanded and cultivated for subsequent experimental detection.

### 2.2. Materials and Reagents

A FilmTracer™ LIVE/DEAD^®^ Biofilm Viability Kit (Thermo Fisher, Waltham, MA, USA), Eppendorf tube, blue spearhead, yellow spearhead (Axygen, Union City, CA, USA), Human Heparinase (HPA) kit, folinol reagent, gallic acid, Coomassie brilliant blue R250, o-Nitrophenyl β-D-galactopyranoside (Shanghai Aladdin Biochemical Technology Co., Ltd., Shanghai, China), sheep red blood cells (Qingdao Haibo Biotechnology Co., Ltd., Xi’an, China), crystal violet, Biofilm Metabolic Activity Assay (XTT) kit (Sigma, Munich, Germany), and lemon verbena essential oil: geranial (31.79%), Neral (23.75%), geraniol (22.01%), and D-limonene (10.36%) [18] (Beijing Sulaibao Technology Co., Ltd., Beijing, China) were utilized.

### 2.3. Instruments and Equipment

A microplate reader (MK3 Thermo Fisher, USA), constant temperature incubator (DNP9082 Shanghai Jinghong Instrument Equipment Co., Ltd., Shanghai, China), centrifuge (MIKRO220R Hettich, Westphalia, Germany), micro vibrator (MM-1 Jintan Medical Instrument Factory, Jintan, China), confocal laser scanning microscope (CLSM) (LSM880 Zeiss, Oberkochen, Germany), and biological transmission electron microscope (Tecnai G2 SpiritBiotwin, FEI Company, Hillsboro, OR, USA) were utilized.

### 2.4. Experimental Method

#### 2.4.1. MIC and MBC Determination

The broth microdilution antifungal susceptibility test was used to detect the anti-microbial effect of lemon verbena essential oil on *Pseudomonas*. Shaking culture was performed in the tested bacterial suspension (10^5^ colony-forming units (CFU) mL^−1^) and lemon verbena essential oil (solvent to essential oil is broth MHB) in nutrient broth for 16 h. Essential oil concentrations used were 0.15%, 0.3%, 0.6%, 0.9%, and 1.2%. MIC is defined as the lowest concentration of a drug that inhibits the growth of pathogenic bacteria in the medium. In addition, the bacterial suspension of the test group without visible bacterial growth was cultured on nutrient agar. MBC was defined as the minimum concentration of lemon verbena essential oil without colony growth.

#### 2.4.2. Growth Curve Determination

A total of 1 mL of overnight culture was taken and inoculated in 35 mL of fresh Tryptic Soy Broth (TSB) medium, and stationary culture was performed under different conditions. Sterile TSB medium was taken as a blank control, and 100 μL was taken every 24 h to determine OD600.

#### 2.4.3. Detection of Bacterial Biomembrane Structure by Confocal Laser Scanning Microscope (CLSM)

After 72 h of biomembrane formation, the cover glass was gently rinsed with PBS three times and fixed in 4% polyformaldehyde (PFA) at 4 °C for 4 h, rinsed with PBS three times, and then stained with the bacterial live/dead staining membrane tracer TM live/dead™ Biofilm Activity Kit (Film Tracer™, molecular probe*, Thermo Fisher) in the dark for 30 min. The cover glass was rinsed three times with PBS to remove excess stains. Finally, the biomembrane was observed and photographed with 63×/1.40 oil lens of CLSM under 488 nm excitation.

#### 2.4.4. Determination of Extracellular Polysaccharide Content

Overnight bacterial cultures were inoculated in LB broth (1 × 10^5^ CFU/mL) with different concentrations of essential oils (0 MIC, 0.5 MIC, 1 MIC, 2 MIC, 4 MIC) and cultured at 37 °C for 9 h. After centrifugation at 10,000 rpm for 1 min, 1 mL of the collected supernatant and 1 mL of PBS (containing 10 mg/mL azurol skin powder HPA) were cultured at 37 °C for 2 h. After adding 2 mL of 10% (*v*/*v*) trichloroacetic acid to stop the reaction, the EPS content and inhibition rate were quantified by recording the optical density at 600 nm [19].

#### 2.4.5. Cell Membrane Permeability Test

Using O-nitrophenyl p-galactopyranoside (ONPG, Sigma, St. Louis, MO, USA) as a substrate, the β-gal activity in *Pseudomonas D4* was measured to determine the intimal permeability. The active culture of the *Pseudomonas D4* strain grown was rinsed twice with PBS, and then the culture was diluted in the same buffer to obtain OD600 absorbance values between 0.5 and 0.7. An untreated bacterial solution was used as a control (i.e., 0 MIC group). The OD_600_ nm was approximately 0.2. The specimens were cultured at 28 °C for 8 h, and then 10 mL of cell suspension was taken from each specimen and centrifuged at 4 °C and 8000× *g* for 15 min. The β-gal activity was determined through the addition of 4 mL of 0.05 mol/L ONPG substrate dissolved in 0.1 M phosphate buffer to each tube containing 0.2 mL of cell supernatant and 0.8 mL of Z buffer (0.06 M Na_2_HPO_4_, 0.04 M NaH_2_PO_4_·H_2_O, 0.01 M KCl, and 0.001 M MgSO_4_·7H_2_O). The solution was allowed to react at 37 °C for 30 min, and then the reaction was stopped by adding 0.5 mL of 1 M Na_2_CO_3_. Finally, the absorbance value was measured at 420 nm in a microplate reader (Gen 52.06, Synergy HT, Biotech, VT, USA).
β-Galactosidase activity (U/mL) = (OD_420_nm × A)/(B × C × 0.0045)
where A is the reaction system (mL), 5 mL; B is the reaction time (min), 40 min; C is the volume of sample used in analysis (mL), 10 mL; and 0.0045 (mL/n mol) is the extinction coefficient.

#### 2.4.6. Hemolysis Test

*Pseudomonas D4* was cultured in brain–heart infusion (BHI) supplemented with 0, 0.5 MIC, 1 MIC, 2 MIC, or 4 MIC of essential oils at 28 °C with shaking (130 g) for 18 h. After incubation, the culture was centrifuged (4000× *g*, 4 °C, 10 min) and then added into the mixture containing 900 μL of hemolysin buffer and 100 μL of freshly washed sheep red blood cells. The supernatant of essential oil was used as the negative control (0% hemolysis), while the positive control (100% hemolysis) was prepared through the addition of 250 μL of 1% (*v*/*v*) Triton X-100 into hemolysin buffer and sheep red blood cells. The specimens were cultured at 37 °C for 30 min, and then centrifuged at 4000× *g* at 4 °C for 10 min. A microplate reader was used to collect the OD_450_ reading of the obtained supernatant. The percentage of hemolysis was determined by comparing the sample with the positive control.

#### 2.4.7. Biomembrane Formation and Metabolic Activity Experiment (XTT Experiment)

Seventy-two hours after biomembrane formation, the cover glass was rinsed with PBS three times to remove the suspended cells, fixed with methanol at room temperature for 30 min, and then dried at room temperature. Afterwards, the sample was dyed with 200 μL of 0.1% crystal violet solution for 30 min. The cover glass was rinsed with PBS three times and air dried or dried at 65 °C. Finally, 200 μL of 95% ethanol was added to release crystal violet, and the absorbance was measured by a microplate reader at 595 nm.

Seventy-two hours after biomembrane formation, 2000 μL of fresh medium was added to each well (the final concentrations were 0, 0.5, 1, 2, and 4 MIC) and incubated at 28 °C for 8, 16, and 24 h. XTT labeling reagent and electronic coupling reagent in an XTT kit were melted in a 37 °C water bath, and then the two reagents were mixed in a ratio of 50:1 with XTT reagent. Then, 100 μL of PBS and 50 μL of XTT were added to each well and incubated for 24 h at 28 °C away from light. Finally, the absorbance was detected at 450 nm.

#### 2.4.8. Biomembrane Integrity Test

*Pseudomonas D4* was inoculated overnight and subcultured with a dilution of 1:200 (final volume 2000 μL) at 28 °C in a 24-well cell culture plate paved with climbing tablets. The specimen was continuously cultured for 24 h, and then the supernatant was removed and replaced with fresh LB culture medium three consecutive times. Then, 72 h after biomembrane formation, 2000 μL of fresh medium was added to each well (the final concentrations were 0, 0.5, 1, 2, and 4 MIC), and incubated at 28 °C for 8, 16, and 24 h. Lemon verbena essential oil at different concentrations was added into the bacterial solution.

Based on the method used by Chen et al. [20], the concentration of *Pseudomonas D4* was adjusted to 1 × 10^5^ CFU/mL and centrifuged at 4000× *g* for 4 min. After rinsing twice with distilled water, the particles were suspended with sterile PBS buffer, and then the suspension was incubated for 24 h. Following this, 100 μL of bacterial suspension was removed from the 96-well plate at different times and detected by spectrophotometry at 260 nm. In addition, specimens with the same concentration without bacteria and solutions containing only bacteria without specimens were used as controls. The specimens treated with cefixime were used as positive controls. The amount of DNA released by the bacteria was detected. Then, 100 μL of Coomassie brilliant blue R250 was added to the 96-well plate and cultured in the dark for 3 min. The released protein was detected by absorption spectrophotometry at 280 nm.

#### 2.4.9. Observation Experiment of Bacterial Structure by Transmission Electron Microscopy (TEM)

Different concentrations (0, 0.5 MIC, 1 MIC, 2 MIC, or 4 MIC) of lemon verbena essential oil were added to the initial suspension, and the bacterial cells were obtained by centrifugation after shaking the culture at 30 °C for 6 h. The specimens were fixed with 2.5% glutaraldehyde, rinsed with 0.01 mol L^−1^ phosphate buffer, and fixed in 1% osmium tetroxide for 2 h, and then rinsed again. Afterwards, the specimens were continuously dehydrated with ethanol (30%, 50%, 70%, 80%, 90%, 95%, and 100%) and treated with pure acetone for 20 min. The specimens were embedded in epoxy resin (Spurr) for polymerization, and then cut into thin slices with an ultra-fine microtome and double-stained with uranyl acetate and lead citrate. Finally, TEM images were obtained by transmission electron microscopy [21].

### 2.5. Statistical Analysis

Multiple comparisons were performed by one-way analysis of variance (ANOVA) using SPSS 22.0, and the results were expressed as means ± standard deviation.

## 3. Experimental Result

### 3.1. MIC and MBC

After the culture from the bacterial solution had been treated with lemon verbena essential oil at different concentrations, the OD_600_ absorbance of the culture was measured. From the clarification and absorbance of the bacterial solution, there was no significant difference between the bacterial solution added with 0.3% essential oil and the control group. There was no bacterial growth in the plate cultured and coated with the bacterial solution added with 0.6%, 0.9%, and 1.2% essential oil. Therefore, it was determined that the MIC and MBC of lemon verbena essential oil on the *Pseudomonas D4* strain isolated from *Pseudosciaena* were 0.3% and 0.6%, respectively.

### 3.2. Growth Curve

In order to analyze the antibacterial activity of lemon verbena essential oil, the growth curves of *Pseudomonas D4* under the action of different concentrations of lemon verbena essential oil were drawn. As shown in Figure 1A, the growth rate of *Pseudomonas* was relatively faster in the control group without the treatment of lemon verbena essential oil; it then entered a logarithmic growth period after 10 h and a stable period after 40 h. The growth of *Pseudomonas* treated with 2 × MIC lemon verbena essential oil was completely inhibited.

### 3.3. Cell Membrane Integrity

The integrity of the cell membrane can be assessed by detecting the leakage of nucleic acids and proteins in the external environment. Nucleic acids and proteins are foundational substances throughout the bacterial cell membrane and cytoplasm, and they flow out when the bacterial membrane is disrupted. In order to elucidate the antibacterial effect of lemon verbena essential oil on *Pseudomonas D4*, its effects on the permeability and integrity of the *Pseudomonas D4* cell membrane were investigated. After the destruction of the cell membrane caused by the lemon verbena essential oil, small molecular ions flowed out first, followed by nucleic acids, proteins, and other macromolecular substances [22]. This phenomenon can be verified by measuring the change in cell integrity, which is characterized by the leakage of nucleic acid and protein through the bacterial membrane at 260 nm and 280 nm [23], respectively. According to the absorbance (OD 260 nm) of the culture filtrate of *Pseudomonas D4* cells exposed to different concentrations of lemon verbena essential oil, the absorbance value increased with the extension of the culture time (Figure 1B). Meanwhile, the 260 nm absorbance value increased significantly with the increase in the lemon verbena essential oil concentration (*p* < 0.05). After the treatment of lemon verbena essential oil at different concentrations (0 × MIC, 1/2 × MIC, 1 × MIC, 2 × MIC, and 4 × MIC), the absorbance values of *Pseudomonas D4* at 260 nm were 0.02, 0.23, 0.55, 0.78, and 0.89, respectively (Figure 2A). As the main signal of membrane destruction and cell lysis, the release of protein at 260 nm also requires monitoring [24]. The protein concentration after treatment with lemon verbena essential oil at different concentrations is shown in Figure 1C. Similar to the results of the nucleic acid release, the protein content also increased with the extension of the culture time. The 280 nm absorbance value increased significantly with the increase in the concentration of lemon verbena essential oil (*p* < 0.05). The results of cytoplasmic leakage showed that lemon verbena essential oil could induce the rupture of the bacterial cell membrane. Therefore, the bactericidal effect of lemon verbena essential oil can be inferred by destroying the cell integrity of *Pseudomonas D4* cells.

### 3.4. Biomembrane Biomass Determination and Biomembrane Metabolic Activity Test (XTT Reduction Test)

The anti-biomembrane activity of lemon verbena essential oil can be verified by measuring the changes in the biomembrane biomass using crystal violet staining. As shown in Figure 2A, the content of *Pseudomonas D4* showed a downward trend with the extension of the culture time. The concentration of essential oil increased and the content of *Pseudomonas D4* decreased, indicating that the utilization of lemon verbena essential oil not only destroyed the biomembrane, but also inhibited its production. The greater the utilization of essential oil, the better the inhibition effect. The results obtained by the XTT reduction experiment were consistent with those obtained using crystal violet staining.

There was a significant difference between the experimental group treated with essential oil and the control group (*p* < 0.05). Compared with the control group, the lemon verbena essential oil at different concentrations significantly reduced the activity of the cell metabolism in the biomembrane. These results showed the obvious role of lemon verbena essential oil in preventing the formation of the biomembrane and in the destruction of the biomembrane. The reported theories on the destruction of the biomembrane include the following. (i) Permeation limitation theory. Extracellular polymers produced by microorganisms themselves surround the biomembrane cells, forming a natural barrier to prevent the penetration of antibacterial agents. Meanwhile, the penetration of antibacterial agents can be prevented through the high density of microorganisms, small gaps between bacteria, and a large volume of hydrophobic proteoglycans in the biomembrane [25]. (ii) Nutrition restriction theory. The majority of the bacteria are insensitive to antimicrobial agents due to the limited supply of nutrients in the biomembrane [26]. (iii) Other theories include the degradation or modification of antibacterial agents and the enzymes produced against antibacterial agents [27]. As shown in Figure 2B, XTT detection was used as an indicator attached to living cells. After 8 h of treatment, the metabolic activity of the lemon verbena essential oil experimental group decreased significantly, but decreased slightly after 16 h and 24 h of treatment.

### 3.5. Determination of EPS

The accumulation of EPS thickens the biofilm and enhances the resistance of bacteria to antibiotics and other undesirable factors. Therefore, the decrease in the production of EPS could benefit the inhibition of *Pseudomonas D4* [28]. The effect of lemon verbena essential oil on the intervention and inhibition rate of bacterial cell EPS during the formation of the biomembrane was measured. As shown in Figure 2C,D, with the increase in the concentration of lemon verbena essential oil, the EPS content in the biomembrane of *Pseudomonas D4* decreased significantly (*p* < 0.05), while the inhibition rate increased significantly (*p* < 0.05), indicating that the secretion of EPS by *Pseudomonas* cells could be inhibited by lemon verbena essential oil during the formation of the biomembrane. The protection of the polymer matrix outside the biomembrane makes it difficult to eliminate them once formed in foodborne pathogens in a food-processing environment or in food [29]. The extracellular polymer content determines the integrity of the biomembrane structure and protects bacterial cells from the invasion of antibacterial agents [30]. EPS is the major component of the polymeric extracellular matrix in the biomembrane, the production of which can be inhibited by lemon verbena essential oil during the formation of *Pseudomonas D4*, as observed in this study. The reduction in EPS content led to the loose structure of the biomembrane, making it easier for the bacterial cells in the biomembrane to be inactivated by antibacterial agents, which is conducive to the inhibition of EPS production through antibacterial agents in the food industry [31]. The decreased production of EPS impacts the formation of biofilms, as EPS has a vital role in the early stage of biofilm formation [32].

### 3.6. Cell Membrane Permeability Test

β-Galactosidase (β-Gal) can be used as a cell osmotic marker enzyme to evaluate the permeability damage of cell membranes caused by lemon verbena essential oil. In lemon verbena essential oil and *Pseudomonas D4* suspension at different concentrations, β-Gal activity increased with the increase in the concentration and time, and was about 1.29, 1.35, 1.40, and 1.50 times higher than that of CK at 0.5 MIC, 1 MIC, 2 MIC, and 4 MIC for 8 h, respectively (Figure 2E). The β-Gal activity increased rapidly within the first hour after the addition of the lemon verbena essential oil at different concentrations, and then gradually increased with the extension of the incubation time. Lemon verbena essential oil could destroy the cell membrane of *Pseudomonas D4* and increase its permeability, resulting in the leakage of intracellular components.

### 3.7. Hemolysis Test

After the incubation of lemon verbena essential oil at different concentrations in sheep red blood cells, the hemolysis rate decreased with the increase in the lemon verbena essential oil concentration (Figure 3A,B). Hemolysis promotes the release of hemoglobin through the instability of the cell membrane, leading to the dysfunction of important organs such as the kidneys, liver, and heart [33]. Lipophilic molecules in essential oil can interact with the erythrocyte membrane to increase its fluidity. However, this interaction destroys the cell membrane, leading to the disordered inflow of ions and water, and eventually erythrocyte lysis [34]. A hemolysis test can determine how the components of the essential oil interact with human biological entities (red blood cells) at the cellular level; therefore, it can be used as one of the indicators of cytotoxicity. Despite the strong antibacterial ability, some essential oils will lead to a higher hemolysis rate at low concentrations, making it difficult to use as an antibacterial agent. For instance, Rodrigues et al. [35] identified the potential application value of essential oil in the control of Leishmania amazonensis. However, its hemolysis rate could be as high as 50% under a concentration of 50 μL/mL. Therefore, lemon verbena essential oil cannot cause hemolysis, despite its strong antibacterial effect.

### 3.8. Detection of Bacterial Structure by Confocal Laser Scanning Microscope (CLSM)

TSB medium containing *Pseudomonas D4* (10^6^ CFU/mL) was added to the sterile cover glass to form a biomembrane. The biomembrane cells attached to the cover glass, being treated with lemon verbena essential oil at different concentrations, were labeled with 4′,6-diamidino-2-phenylindole (DAPI) (10 μg/mL) and kept away from light for 10 min. Untreated specimens were used as a control. The stained biomembrane was observed by CLSM at an excitation wavelength of 364 nm and an emission wavelength of 454 nm [36].

Figure 4 shows the untreated control group (CK) and *Pseudomonas D4* cells exposed to 1/2 × MIC, 1 × MIC, 2 × MIC, and 4 × MIC lemon verbena essential oil. In the CLSM analysis, the reduction in the fluorescence region indicated the reduced number of bacteria in the biomembrane [37]. As shown in Figure 4 CK, most cells in the *Pseudomonas D4* control group were alive (green), and all cells were single cells without visible aggregates. In contrast, most of the treated cells appeared as green aggregates (living cells) (Figure 2C–Figure 4 1/2 × MIC, 1 × MIC), which could be attributed to the aging of the cells since the cells were treated with the antibacterial lemon verbena essential oil in the early stable period, during which the cells showed the greatest resistance to adverse conditions. Initially, the biomembrane had a dense and complete three-dimensional structure, which was destroyed to varying degrees after being treated with lemon verbena essential oil at different concentrations. With the increase in the lemon verbena essential oil concentration, the three-dimensional structure of the biomembrane completely collapsed, and only a small number of bacterial cells adhered to the glass (Figure 4 1/2 × MIC, 1 × MIC).

A reason for this may be that lemon verbena essential oil is hydrophobic, which can increase the hydrophobicity of cells, thus making it conducive to the adhesion between cells. Yaron et al. [38] found that the hydrophobic components of essential oil could destroy the cytoplasmic membrane, leading to protein leakage, thereby inducing cell aggregation, which may be due to the loss of transmembrane transporters such as SecA2 [39]. Lemon verbena essential oil can interact with the cytoplasmic membrane, resulting in damage to its structure and membrane proteins (such as SecA2). The CLSM results showed a favorable scavenging effect of lemon verbena essential oil in the *Pseudomonas D4* biomembrane.

### 3.9. Observation of Bacterial Structure by TEM

As shown in Figure 5A, the TEM image showed that the external morphological characteristics of *Pseudomonas D4* changed after the treatment of lemon verbena essential oil at different concentrations (0 × MIC, 1/2 × MIC, 1 × MIC, and 2 × MIC). Untreated *Pseudomonas D4* cells retained their original morphology (Figure 5A CK). *Pseudomonas D4* cells treated with lemon verbena essential oil underwent lysis, resulting in the release of their cell contents into the surrounding environment, and therefore became empty. Compared with CK, cell fragments appeared around the damaged cells in the treatment group, revealing electron translucent cytoplasm. The cell wall and membrane of the CK were intact, and the peptidoglycan layer and plasma membrane were well maintained. The cracked cells were observed in *Pseudomonas D4* cells treated with essential oil, with destroyed cell walls and cell membranes, and locally isolated cell membranes and cell walls. In addition, the cell degradation was accompanied by the outflow of cytoplasm from the damaged cells. He et al. [40] found that *Staphylococcus aureus* had a regular and smooth surface in the control group. *Staphylococcus aureus* was seriously damaged after the treatment with Atractylodes lancea rhizomes essential oil, revealing an irregular shape, and some cells even underwent lysis.

### 3.10. Anti Quorum Sensing (QS) Test

The purple stain bacilli QS system was used for determination. It is known that the production of violetin (a purple pigment) is controlled by QS in wild-type violet *C. violaceum* CECT 494, so as to correspond to the synthesis of the autoinducer of N-acyl-t-homoserine lactones (AHLs), such as C6-acyl homoserine lactones and C4-acyl homoserine lactones. For mutants of violet *C. violaceum* CV026 and VIR07, violetin must be produced by the addition of an exogenous autoinducer (short-chain AHLs and long-chain AHLs, respectively). The well diffusion method was used to detect the anti-QS activity of lemon verbena essential oil [41]. In this test (Figure 5B), a clear halo was formed by the inhibition of bacterial growth, while a turbid halo containing achromatic bacterial cells was formed as the positive result of QS inhibition.

## 4. Conclusions

Taking *Pseudomonas* D4 as the research object, the destruction effect and mode of action of lemon verbena essential oil on its cells were studied. After *Pseudomonas* D4 was treated with lemon verbena essential oil, the absorbance values of OD260 and OD280 increased, which destroyed the cell integrity of *Pseudomonas* D4. Lemon verbena essential oil significantly reduced the activity of cell metabolism in biofilm and played an obvious role in preventing the formation and destruction of biofilm. The content of extracellular polysaccharide decreased significantly with the increase in the concentration of lemon verbena essential oil, and the inhibition rate increased significantly with the increase in the concentration of essential oil, indicating that lemon verbena essential oil can inhibit the secretion of extracellular polysaccharide of *Pseudomonas* D4 during the formation of biofilm. The cells treated by lemon verbena essential oil appeared to have depressions and folds, and the cell content flowed out to form a transparent cavity. Lemon verbena essential oil destroyed the cell wall, increased the permeability of the cell membrane, and the cell content flowed out, which led to cell death.

## Figures and Tables

**Figure 1 molecules-28-03102-f001:**
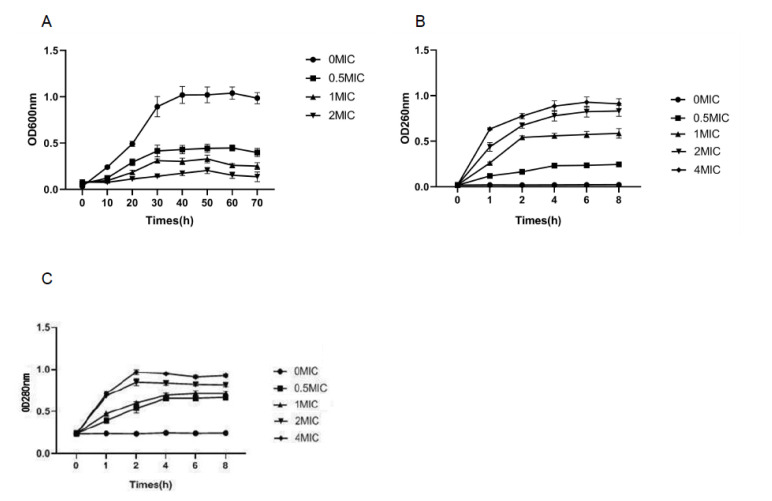
Effects of different amounts of lemon verbena essential oil on *Pseudomonas D4*. (**A**) Growth curve of *Pseudomonas D4* under different concentrations of lemon verbena essential oil. (**B**) Absorbance of *Pseudomonas D4* exposed to different concentrations of lemon verbena essential oil at 260 nm. (**C**) Absorbance of *Pseudomonas D4* exposed to different concentrations of lemon verbena essential oil at 280 nm.

**Figure 2 molecules-28-03102-f002:**
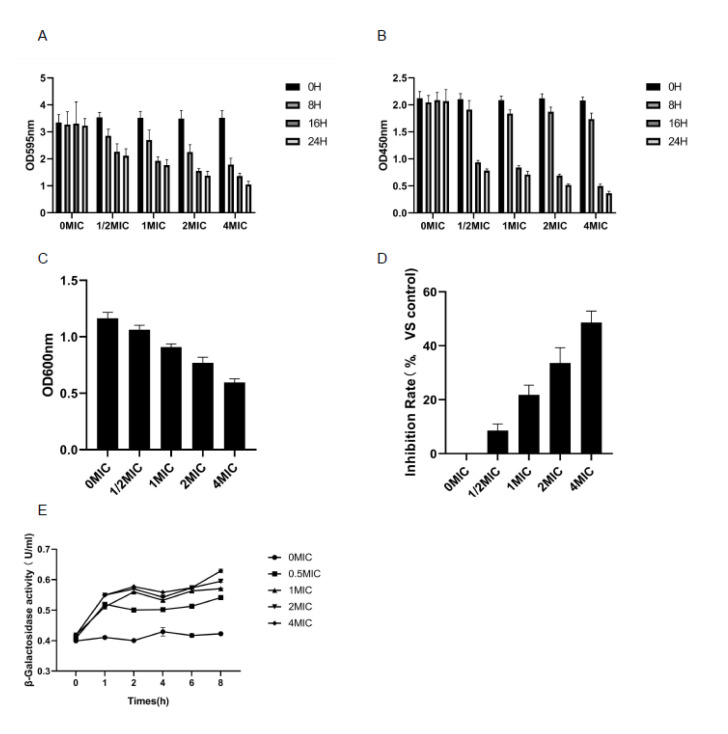
Disruption of *Pseudomonas D4* cell membrane and biofilm by lemon verbena essential oil. (**A**) Determination of biofilm biomass (crystal violet method) of *Pseudomonas D4* in different times and different concentrations of lemon verbena essential oil at 595 nm. (**B**) Biofilm metabolic activity assay (XTT reduction assay) of *Pseudomonas D4* in different times and different concentrations of lemon verbena essential oil at 450 nm. (**C**) Absorbance of *Pseudomonas D4* treated with different concentrations of lemon verbena essential oil (0 × MIC, 1/2 × MIC, 1 × MIC, 2 × MIC, and 4 × MIC) at 600 nm. (**D**) Exopolysaccharide inhibition rate of *Pseudomonas D4* treated with different concentrations of lemon verbena essential oil (0 × MIC, 1/2 × MIC, 1 × MIC, 2 × MIC, and 4 × MIC). (**E**) Endomembrane permeability test of *Pseudomonas D4* treated with different concentrations of lemon verbena essential oil (0 × MIC, 1/2 × MIC, 1 × MIC, 2 × MIC, and 4 × MIC).

**Figure 3 molecules-28-03102-f003:**
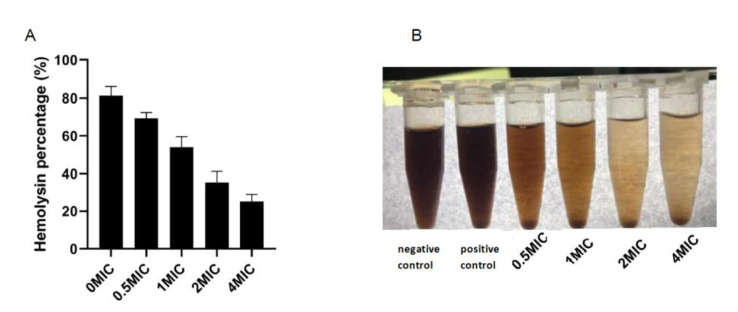
Lemon verbena essential oil inhibits the hemolytic effect of *Pseudomonas D4* on sheep red blood cells. (**A**) Hemolysis assay of *Pseudomonas D4* treated with different concentrations of lemon verbena essential oil (0 × MIC, 1/2 × MIC, 1 × MIC, 2 × MIC, and 4 × MIC). (**B**) Hemolysis assay of *Pseudomonas D4* treated with different concentrations of lemon verbena essential oil (0 × MIC, 1/2 × MIC, 1 × MIC, 2 × MIC, and 4 × MIC).

**Figure 4 molecules-28-03102-f004:**
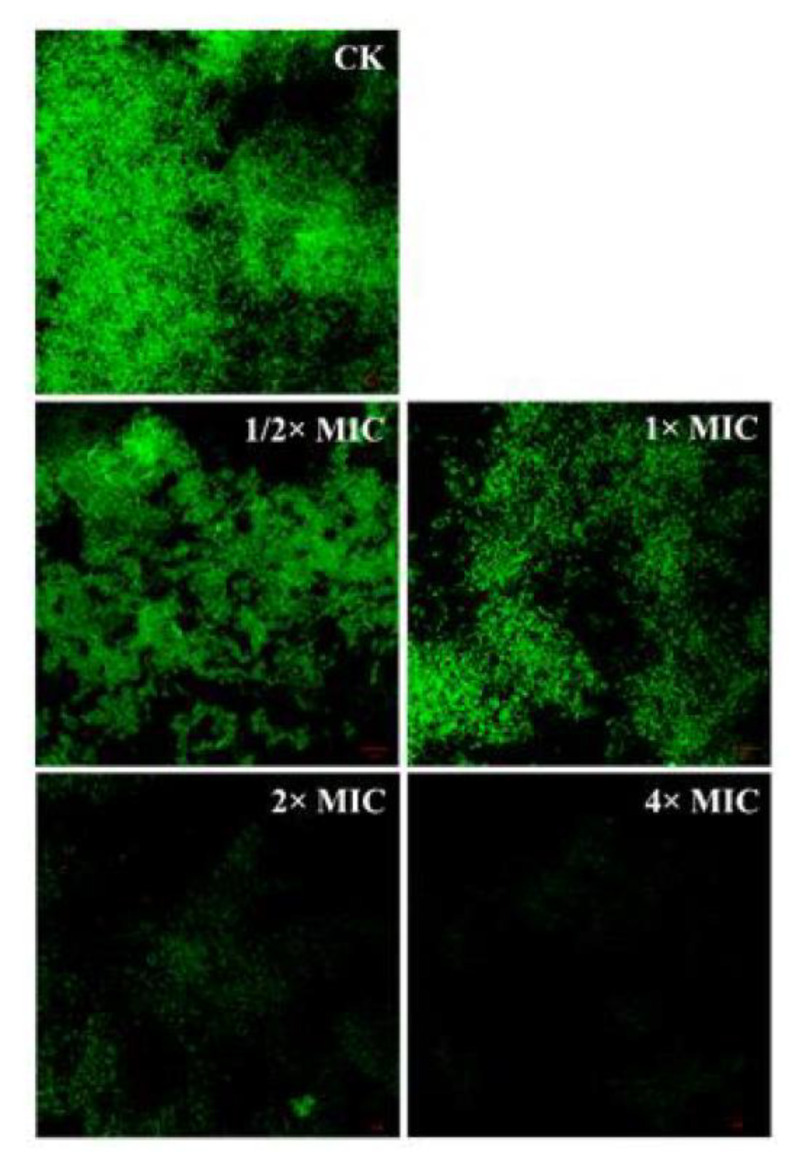
Inhibitory effect of lemon verbena essential oil on *Pseudomonas D4* biofilm. CLSM photo of *Pseudomonas D4* treated with different concentrations of lemon verbena essential oil (0 × MIC, 1/2 × MIC, 1 × MIC, 2 × MIC, and 4 × MIC).

**Figure 5 molecules-28-03102-f005:**
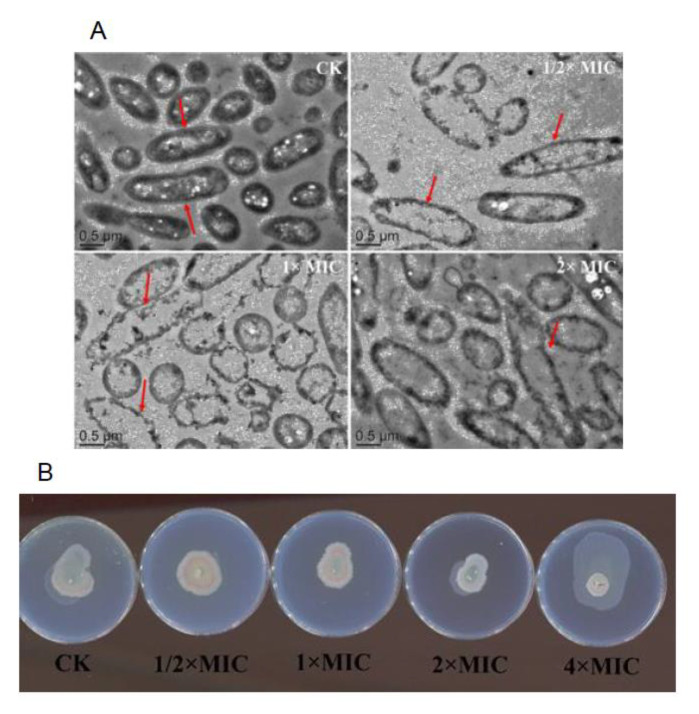
Effect of lemon verbena essential oil on *Pseudomonas D4* under transmission electron microscopy. (**A**) Transmission electron microscope images of *Pseudomonas D4* treated with different concentrations of lemon verbena essential oil (0 × MIC, 1/2 × MIC, 1 × MIC, and 2 × MIC). (**B**) Anti-QS assay of *Pseudomonas D4* treated with different concentrations of lemon verbena essential oil (0 × MIC, 1/2 × MIC, 1 × MIC, and 2 × MIC).

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
