# Peer review of "Antibacterial Activity and Antibacterial Mechanism of Lemon Verbena Essential Oil"

_molecules, 2023, doi:10.3390/molecules28073102_

Round 1

Reviewer 1 Report

Dear Authors,

The research related to antimicrobial is still interesting topic including using the essential oils which was reported as a good antimicrobial agent. It is important to check the quality of oils since when we collected at different time and also different isolation method will gave different components. Please add the component of the essential oil, such as by using GC-MS analysis.

Pseudomonas D4 is interesting microbe to be used as the microorganism test, but it is important to give the detail determination of the Pseudomonas D4, perhaps by using previous publication using the same Pseudomonas D4.

Data analysis is an important part on the research, would you please add the data analysis in the methods.

Some typos please revise and the others comments please see in the manuscript.

Reviewer 2 Report

1.       Even though the data presented are interesting. The novelty in this manuscript is compromised in some subtopics. Introduction, the motivation/justification is confusing. Results need attention, e.g. integration. The literature review appears to be exhaustive. However, in some subtopics, the discussion is absent. While others exhibit a weak argument, and few have a strong discussion.

2.       Species names are in an italic case.

3.       Some advice is in the main text

4.       References need a lot of attention.

5.       The manuscript is poorly written, and the understanding of the ideas exposed by the authors is compromised several times during the whole manuscript. I would recommend a complete review of the text by a native English speaker before any attempt to re-submit this manuscript.

The manuscript exhibited a weakness. The title, the aim of the abstract, and the aim of the introduction have not congruent. Trends and new perspectives, gaps, conflicts, the problem or new problems that arise, and discussion in the new lines of research that currently exist and are emerging are confusing. Perhaps, result integration will improve the edition of the manuscript, also with figures or tables that contain the important milestones of these topics.

Example:

Data integration:

MIC and MBC are values to categorized an organism as susceptible, intermediate, resistant or nonsusceptible.

Material and methods.

Example

In the experimental design, the following factors are absent: reference bacterial strains, antibiotic-control (derived from plants), and the resistance scheme of Pseudomonas D4 to conventional antibiotics.

The authors do not indicate the solvent to essential oil. To determine MIC by dilution methods, antibiotics are also needed in a substance that require preliminary dissolution to obtain a stock solution and then dilution to obtain an appropriate starting concentration. For most antibiotics, water is both a solvent and a diluter, including for most beta-lactams, fluoroquinolones and aminoglycosides. Some require alcohol as a solvent, especially macrolides, chloramphenicol and rifampicin, while others require a phosphate buffer or dimethyl sulfoxide (DMSO). Dissolved and diluted antibiotics are used to make working solutions in broth or agar cultures.

On the other hand, to determine MIC values, all quantitative methods use Mueller–Hinton (MH) medium either in the form of agar (MHA) or broth (MHB), in some cases additionally supplemented with, for example, 5% lysed horse blood or other compounds depending on bacteria or antibiotic type

Results

Example

1)      Now, MIC or MBC values are sufficient and acceptable evidence to indicate bacterial susceptibility. On manuscript Response-concentration plots are irrelevant and take up space for relevant results.

2)      It misses gas/mass chromatographic data of essential oil

3)      Also, it misses the molecular determination of Pseudomonas D4
